# Pulsed ultrasound promotes secretion of anti-inflammatory extracellular vesicles from skeletal myotubes via elevation of intracellular calcium level

Atomu Yamaguchi[1], Noriaki Maeshige[1]*, Hikari Noguchi[1], Jiawei Yan[2], Xiaoqi Ma[1], Mikiko Uemura[1], Dongming Su[3], Hiroyo Kondo[4], Kristopher Sarosiek[5], Hidemi Fujino[1]

[1]Department of Rehabilitation Science, Kobe University Graduate School of Health Sciences, Kobe, Japan; [2]School of Life Sciences and Technology, ShanghaiTech University, Shanghai, China; [3]Department of Pathology, Nanjing Medical University, Nanjing, China; [4]Department of Health and Nutrition , Shubun University, Ichinomiya, Japan; [5]John B. Little Center for Radiation Sciences, Harvard University T.H. Chan School of Public Health, Boston, United States

*For correspondence:
nmaeshige@pearl.kobe-u.ac.jp

Competing interest: The authors declare that no competing interests exist.

**Abstract** The regulation of inflammatory responses is an important intervention in biological function and macrophages play an essential role during inflammation. Skeletal muscle is the largest organ in the human body and releases various factors which mediate anti-inflammatory/immune modulatory effects. Recently, the roles of extracellular vesicles (EVs) from a large variety of cells are reported. In particular, EVs released from skeletal muscle are attracting attention due to their therapeutic effects on dysfunctional organs and tissues. Also, ultrasound (US) promotes release of EVs from skeletal muscle. In this study, we investigated the output parameters and mechanisms of US-induced EV release enhancement and the potential of US-treated skeletal muscle-derived EVs in the regulation of inflammatory responses in macrophages. High-intensity US (3.0 W/cm$^2$) irradiation increased EV secretion from C2C12 murine muscle cells via elevating intracellular Ca$^{2+}$ level without negative effects. Moreover, US-induced EVs suppressed expression levels of pro-inflammatory factors in macrophages. miRNA sequencing analysis revealed that miR-206-3p and miR-378a-3p were especially abundant in skeletal myotube-derived EVs. In this study we demonstrated that high-intensity US promotes the release of anti-inflammatory EVs from skeletal myotubes and exert anti-inflammatory effects on macrophages.

## eLife assessment

This study illuminates the effects of ultrasound-induced extracellular vesicle interactions with macrophages. It provides **solid** data offering insights that will be potentially **useful** in exploring therapeutic approaches to inflammation modulation, by suggesting that ultrasound-treated myotube vesicles can suppress macrophage inflammatory responses.

## Introduction

Inflammation is a crucial response to defend the body from infection. However, excess and prolonged inflammation can also be harmful and needs to be tightly regulated (*Funes et al., 2018*). Macrophages play a leading role in the innate immune system, causing inflammation during infection

(*Hirayama et al., 2017*). Therefore, regulating the condition of macrophages is a major therapeutic strategy for the control of excess inflammation. Various organs including skeletal muscle are known to release anti-inflammatory factors. Skeletal muscle is the largest organ in the human body accounting for 40% of the body weight (*Trovato et al., 2019*) and responsible for whole-body metabolism, energy homeostasis, and locomotion. Recently, skeletal muscle is attracting attention as a secretory organ of anti-inflammatory factors (*Trovato et al., 2019*), and extracellular vesicles (EVs) are responsible for transporting various factors from skeletal muscle to target organs or cells. EVs are nano-sized vesicles secreted by most types of cells and facilitate cell-to-cell communication (*Trovato et al., 2019*; *Raposo and Stoorvogel, 2013*; *EL Andaloussi et al., 2013*) through transportation of proteins, mRNAs, and miRNAs (*Zomer et al., 2010*), leading to regulation of immune response (*EL Andaloussi et al., 2013*), tissue regeneration (*Bittel and Jaiswal, 2019*), and cell proliferation/differentiation (*Matsuzaka et al., 2016*). EVs from skeletal muscle have also been reported to exert therapeutic effects in various dysfunctional organs (*Rome et al., 2019*; *Bei et al., 2017*; *Madison et al., 2014*) and our previous study revealed that skeletal myotube-derived EVs attenuate inflammatory responses of macrophages (*Yamaguchi et al., 2023*). EVs from mesenchymal stem cells (MSC) also have been reported to suppress inflammatory responses in lipopolysaccharide (LPS)-induced macrophages (*Harrell et al., 2019*; *Xin et al., 2020*; *Shao et al., 2020*) and Kim et al. reported the anti-inflammatory action of MSC EVs depended on the concentration of EVs (*Kim et al., 2019*). Therefore, an enhancement of EV release may enhance the anti-inflammatory effects of muscle EVs.

Ultrasound (US) irradiation is used as a non-invasive therapy and has physiological effects including cell proliferation, suppression of inflammatory signaling (*Ueno et al., 2021*), and increase cell membrane permeability (*Ma et al., 2022*). Our previous study revealed that high-intensity US can promote EV release (*Maeshige et al., 2021*). An increase in intracellular $Ca^{2+}$ concentration is one of the effects of US irradiation, and elevation of intracellular $Ca^{2+}$ is a key factor for EV secretion (*Savina et al., 2003*). Previous studies have reported that low-intensity pulsed ultrasound (LIPUS) promotes EV release from cells (*Deng et al., 2021*; *Li et al., 2023*). Meanwhile, US increases $Ca^{2+}$ influx into cells by increasing cell membrane permeability through sonoporation (*Fan et al., 2010*), but its action has been reported to be dependent on US intensity (*Zeghimi et al., 2015*), so adopting a higher intensity than LIPUS is expected to promote EV release from skeletal muscle cells more efficiently. In addition, stimulus-induced EVs can be altered in their contents and effects compared to EVs released under normal conditions (*Kawanishi et al., 2023*; *Li et al., 2023*), thus EVs released from skeletal muscle by US may have different effects. This study aimed to clarify the intensity dependency of EV release enhancement by US and anti-inflammatory effects of US-induced skeletal muscle-derived EVs on macrophages.

## Results

### US irradiation has no negative effect on myotube viability, protein content, and energy metabolism

To measure the viability of US-irradiated myotubes, MTT assay and Zombie Red immunofluorescence staining were performed. Our results showed no significant decrease in all US groups (*Figure 1A and B*). To assess the potential effect of US irradiation on cultured myotubes, the total protein content in myotubes was measured and no significant difference was observed among all groups (*Figure 1C*). Furthermore, citrate synthase (CS) activity was measured to determine the effect of US on energy metabolism. CS activity in each group did not decrease in the US-treated myotubes (*Figure 1D*).

### US irradiation enhances release of EVs from myotubes

The concentration of EVs in the 3W group was two times higher than that in the control group. The 1W and 2W groups also showed an increase in EVs concentration, which was 1.64 and 1.68 times higher than that in the control group (*Figure 2A*), but no significant difference. Regarding size distribution, the majority of released EVs were in the range of 50–150 nm and the size of EVs did not change in the US groups (*Figure 2B*).

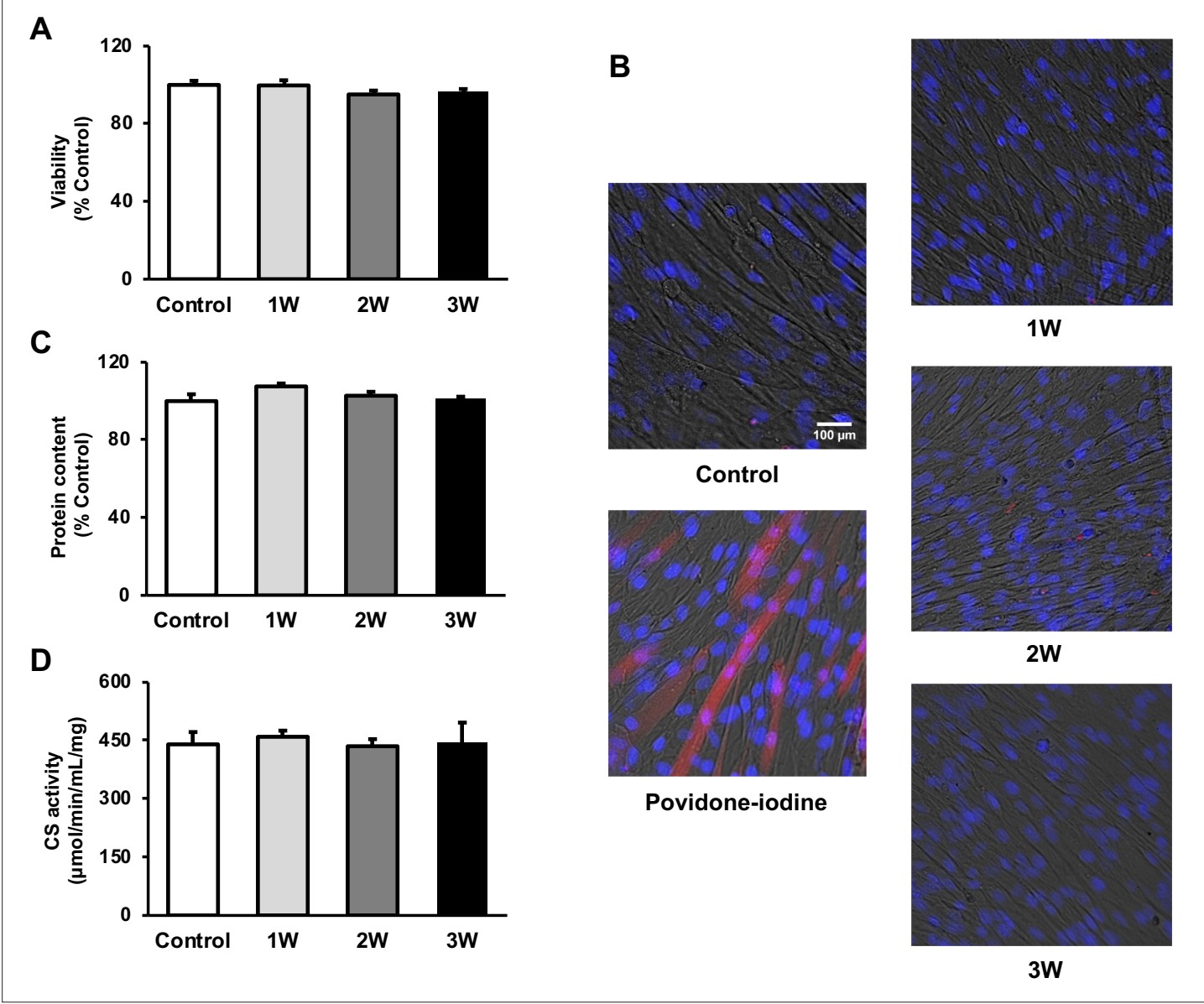

**Figure 1.** The cytotoxicity of ultrasound (US) irradiation on myotubes was investigated. Viability of myotubes was assessed by (**A**) MTT assay and (**B**) Zombie Red staining at 24 hr after US irradiation. (red: Zombie Red, blue: DAPI). (**C**) Total protein content was measured by the Bradford method at 24 hr after US irradiation. (**D**) Energy metabolism in C2C12 myotubes was measured by citrate synthase assay at 24 hr after US irradiation. The US intensities of 1.0 W/cm$^2$, 2.0 W/cm$^2$, and 3.0 W/cm$^2$ were tested. Data are expressed as mean ± SEM. n=4.

## Intracellular Ca$^{2+}$ upregulation mediates US-induced enhancement of EV release from myotubes

To investigate the mechanism of the effect of US on EV release enhancement, we measured intracellular Ca$^{2+}$ levels in myotubes after US irradiation. Compared to the control group, Ca$^{2+}$ level in the 3W group was significantly increased immediately after US irradiation. The 2W group only showed a tendency of increased intracellular Ca$^{2+}$ level, but no significant difference (*Figure 3A*).

Next, we investigated the calcium dependency of the promotive effect of US on EV release using Ca$^{2+}$-free medium. Here, we used the US intensity of 3.0 W/cm$^2$, which showed the greatest EV release enhancement effect. The groups with Ca$^{2+}$-free medium showed significantly lower levels of Ca$^{2+}$ compared with the control and US groups (*Figure 3B*). The concentration of EVs in US group was significantly increased compared to the control group while this increase was not observed in absence

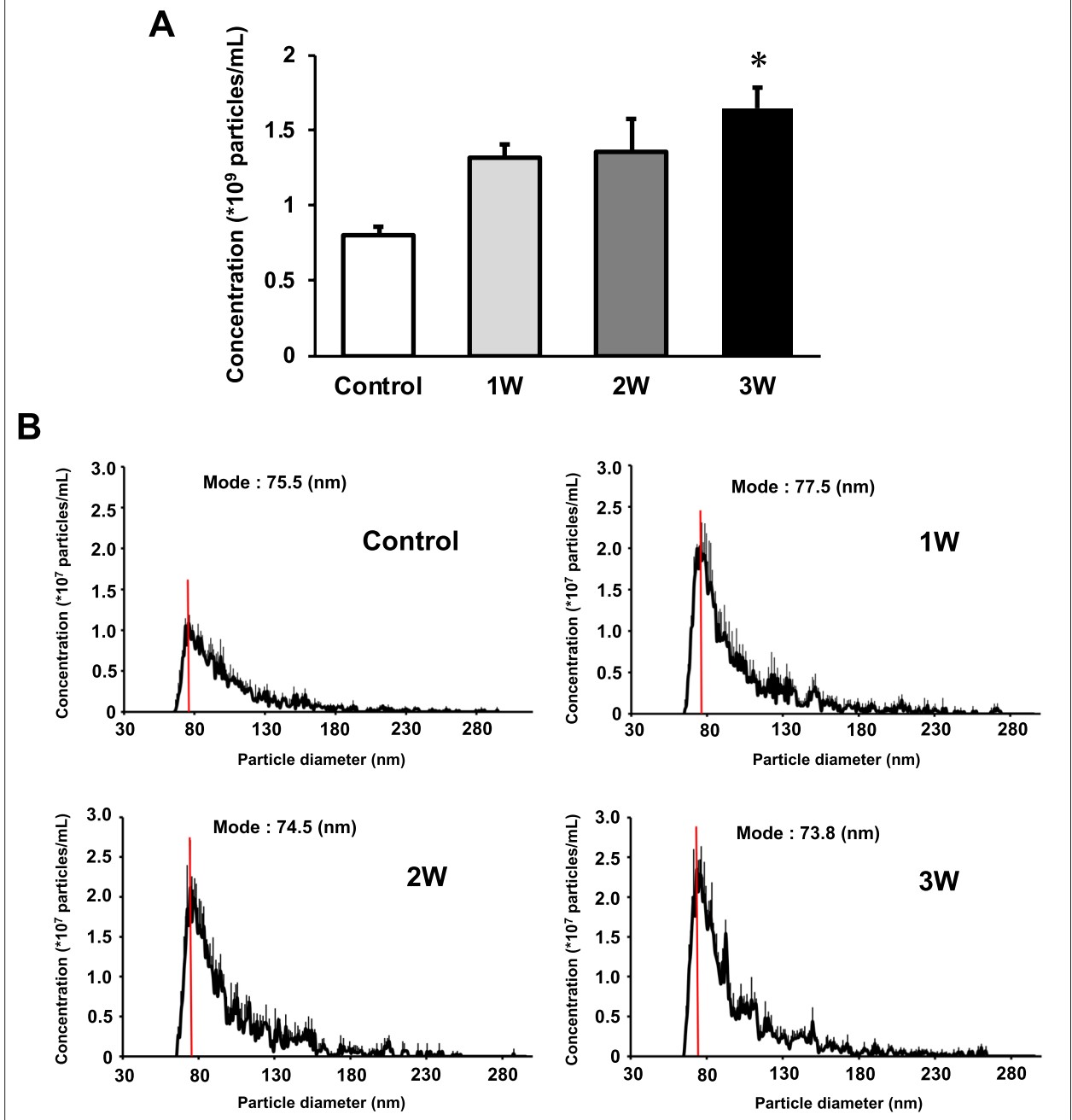

**Figure 2.** Characterization of extracellular vesicles (EVs) from ultrasound (US)-treated/untreated myotubes. EVs were isolated using ExoQuick reagent 12 hr after US irradiation. (**A**) EV concentration in each group was quantified by a qNano system. (**B**) Size distribution of EVs in each group was investigated by a qNano system. The mode value was indicated as a red line. The US intensities of 1.0 W/cm$^2$, 2.0 W/cm$^2$, and 3.0 W/cm$^2$ were tested. Data are expressed as mean ± SEM. *p<0.01, vs. control. n=6.

of Ca$^{2+}$ (*Figure 3C*). Most of the extracted EVs were 50–150 nm in diameter (*Figure 3D*). Myotube viability did not decrease by Ca$^{2+}$-free culture (*Figure 3E*).

## US-induced EVs exert anti-inflammatory effects on macrophages

To investigate the anti-inflammatory effect of myotube EVs on macrophages, qPCR analysis was performed. While C2C12 myotube conditioned medium only showed a tendency to suppress the expression of pro-inflammatory factors, US-treated C2C12 myotube conditioned medium (US-CM) significantly suppressed the expression of pro-inflammatory *Il-1b* and *Il-6*, compared to LPS-stimulation

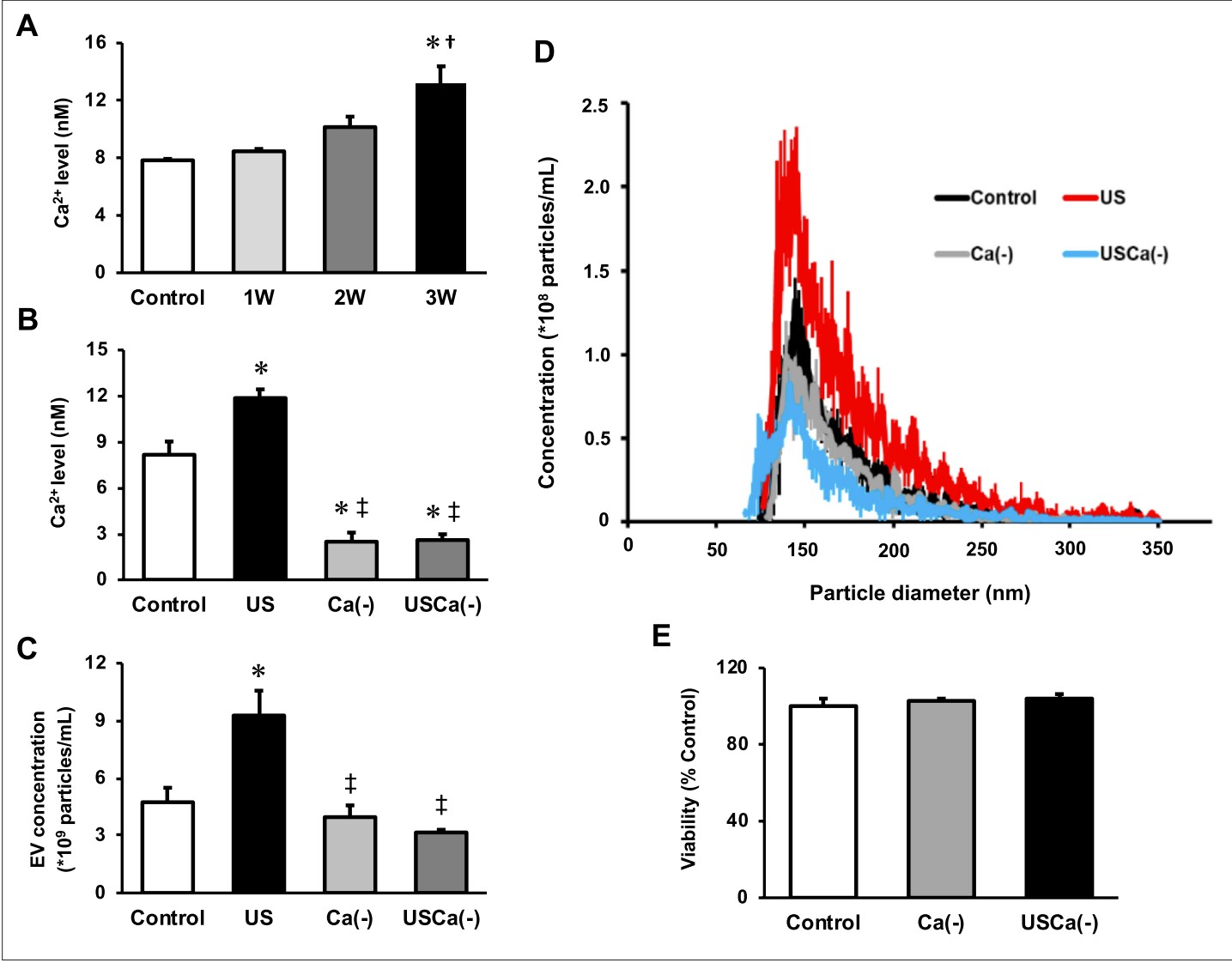

**Figure 3.** Ca²⁺ mediates the promotive effect of ultrasound (US) on extracellular vesicle (EV) release from myotubes. (**A**) Intracellular Ca²⁺ levels were measured after US irradiation. The US intensities of 1.0 W/cm², 2.0 W/cm², and 3.0 W/cm² were tested. (**B**) Cell culture with Ca²⁺-free medium decreased the intracellular Ca²⁺ level and canceled the facilitating effect of US on Ca²⁺ uptake by myotubes. (**C**) Cell culture with Ca²⁺-free medium inhibited the facilitating effect of US on EV release from myotubes. (**D**) Size distribution of EVs in each group. EV concentration and size distribution were quantified by a qNano system. (**E**) Cytotoxicity of cell culture with Ca²⁺-free medium was investigated by MTT assay. Control: untreated; US: 3.0 W/cm² US treatment; Ca(-): Ca²⁺-free culture; USCa(-): 3.0 W/cm² US treatment and Ca²⁺-free culture. Data are expressed as mean ± SEM. *p<0.01 vs. control, †p<0.05 vs 1W, ‡p<0.01 vs. US. n=4.

alone (LPS), and expression of *Il-1b* in US-CM significantly decreased compared to non-treated C2C12 myotube conditioned medium. Furthermore, the expression of *Il-1b* and *Il-6* were not suppressed when the EVs were eliminated from the conditioned media (***Figure 4A***). When EV concentrations were equated, the upregulation of the anti-inflammatory effect of myotube EVs by US was not observed (***Figure 4B***). Myotube EVs did not decrease macrophage viability (***Figure 4C***). EV concentration in each group is shown in ***Figure 4D***.

## miRNA profile change in myotube EVs by US

To investigate the effect of US on miRNA profile in myotube-derived EVs, miRNA sequencing analysis on EVs was performed. A total of 524 miRNAs were identified by proteomic quantitative analysis. Twenty-nine miRNAs were expressed specifically in the control group, and 81 miRNAs were expressed specifically in the US group (***Figure 5A***). Lists of miRNAs specific to each group are shown in ***Supplementary file***

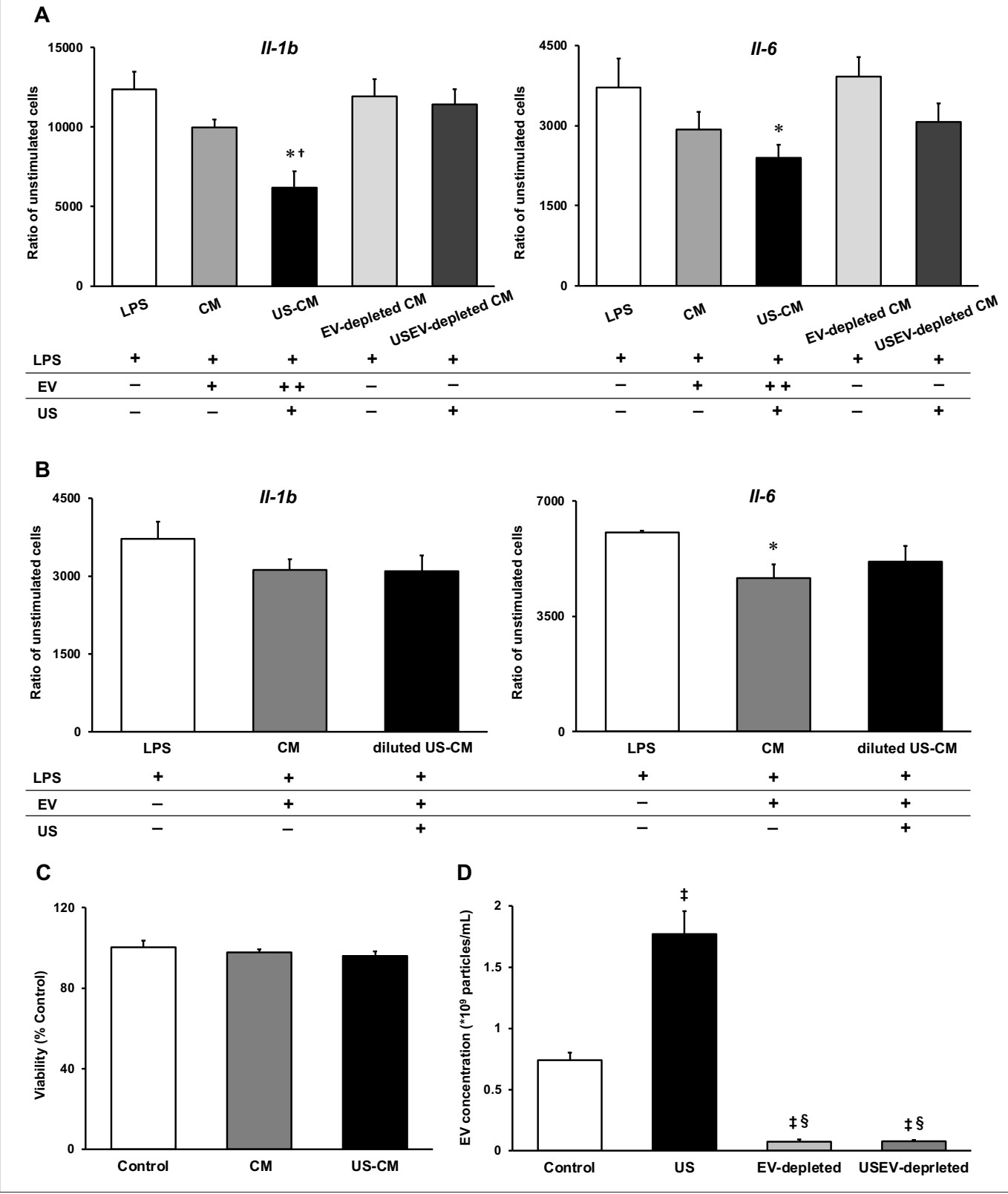

**Figure 4.** Anti-inflammatory effect of extracellular vesicles (EVs) from ultrasound (US)-treated myotubes on bone marrow-derived macrophages (BMDMs). (**A**) The mRNA expression levels of *Il-1b* and *Il-6* were measured by qPCR. Lipopolysaccharide (LPS): LPS-treated BMDMs; CM: BMDMs treated with C2C12 conditioned medium and LPS; US-CM: BMDMs treated with US-irradiated C2C12 conditioned medium and LPS; EV-depleted CM: BMDMs treated with EV-depleted C2C12 conditioned medium and LPS; USEV-depleted: BMDMs treated with EV-depleted C2C12 (US-irradiated)

*Figure 4 continued on next page*

*Figure 4 continued*

conditioned medium and LPS. (**B**) When the concentration of EVs are equated, the enhancement of anti-inflammatory effect of EVs by US was not observed. LPS: LPS-treated BMDMs; CM: BMDMs treated with myotube EVs and LPS; diluted US-CM: BMDMs treated with US-EVs at the same concentration as the EV group and LPS. (**C**) Cytotoxicity of C2C12 conditioned medium on BMDMs was investigated by MTT assay. (**D**) EV concentration in each condition was measured by a qNano system. Data are expressed as mean ± SEM. *p<0.05 vs. LPS, †p<0.01 vs. CM, ‡p<0.01 vs. control, §p<0.01 vs. US. n=4.

*1*. Although miRNAs specific to each group were identified, they accounted for only 0.014% of the total miRNAs, and 99.99% of total miRNAs were common to the control and US groups (*Figure 5B*). According to the standard of a fold change ≥2 or ≤0.5 as well as an FDR <0.05, we screened 13 upregulated miRNAs (*Table 1*) and 14 downregulated miRNAs (*Table 2*) in the EV group versus the control group. Differentially expressed miRNAs are displayed as a volcano plot (*Figure 5C*). The top 10 abundant miRNAs and their proportion to the total miRNAs are shown (*Figure 5D*). In both the control and US groups, the two most abundant miRNAs, miR-206-3p and miR-378a-3p, accounted over 60% of the total.

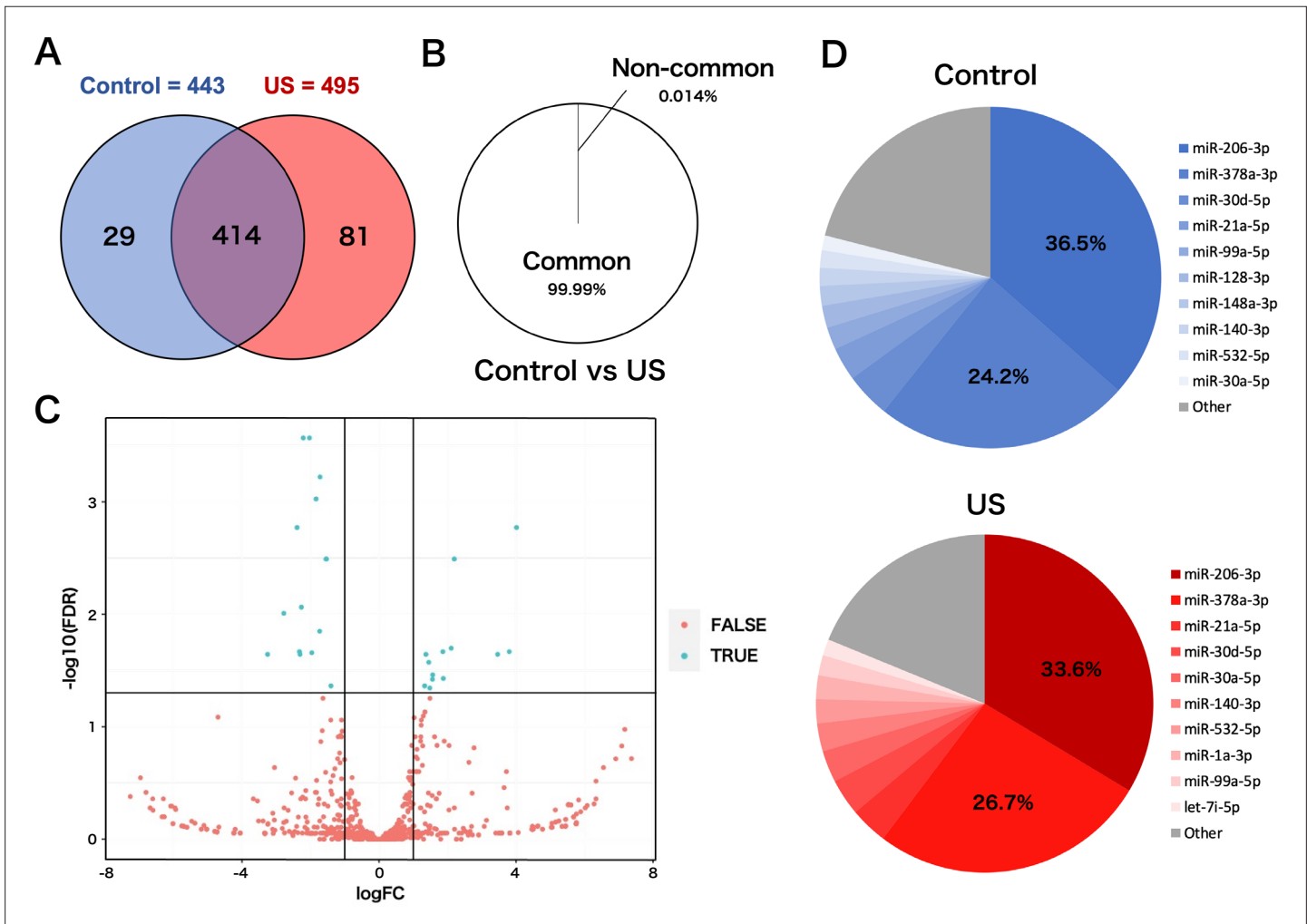

**Figure 5.** miRNA-sequencing analysis in extracellular vesicles (EVs) from ultrasound (US)-treated/untreated C2C12 myotubes. (**A**) miRNA characterization in EVs from US-treated/untreated myotubes. (**B**) Percentage of miRNAs that were common to the control and US groups and those that were not. (**C**) Volcano plot of differentially expressed RNAs in the control group vs. US group. Blue dots represent miRNAs with statistically significant difference and red dots show miRNAs with no statistically significant difference between the control group vs. US group. (**D**) Top 10 abundant miRNAs and their proportion to total miRNA content in each group. n=3.

**Table 1.** Upregulated miRNAs in myotube-derived extracellular vesicles by ultrasound irradiation.

| miRNA | logFC | FDR |
| --- | --- | --- |
| miR-193a-3p | 4.012954262 | 0.000742983 |
| miR-138-5p | 3.800680285 | 0.00141537 |
| miR-223-3p | 3.462174268 | 0.008790418 |
| miR-362-3p | 2.195592968 | 0.009437513 |
| miR-34a-5p | 2.105459261 | 0.009437513 |
| miR-675-3p | 1.876128033 | 0.009966738 |
| miR-106b-5p | 1.862814288 | 0.009966738 |
| miR-30b-5p | 1.566332803 | 0.011740976 |
| miR-188-5p | 1.559409694 | 0.015175451 |
| miR-133b-3p | 1.48161012 | 0.016332293 |
| miR-1a-3p | 1.450599776 | 0.016622751 |
| miR-30a-5p | 1.367918225 | 0.019014654 |
| miR-27b-3p | 1.330089099 | 0.01987041 |

## Discussion

This study demonstrated the facilitatory effect of US irradiation on EV release from cultured skeletal myotubes and the enhanced anti-inflammatory effect of EVs from US-irradiated myotubes. These findings contribute to the development of non-invasive complementary therapy using skeletal muscle-derived EVs for various inflammatory diseases.

Skeletal muscle-derived EVs have been reported to exert therapeutic effects in various dysfunctional organs (*Rome et al., 2019*), including prevention of cardiac damage by ischemia-reperfusion injury (*Bei et al., 2017*), enhancement of neurite outgrowth in motor neurons (*Madison et al., 2014*), and enhancement of angiogenesis (*Ma et al., 2018*). Furthermore, since skeletal muscle EVs mediate cross-talk between skeletal muscle and other tissues (*Whitham et al., 2018*), methods to enhance the

**Table 2.** Downregulated miRNAs in myotube-derived extracellular vesicles by ultrasound irradiation.

| miRNA | logFC | FDR |
| --- | --- | --- |
| miR-128-2-5p | −3.254069169 | 0.000118723 |
| miR-184-3p | −2.779741068 | 0.000118723 |
| miR-615-3p | −2.393817511 | 0.000263853 |
| miR-344d-3p | −2.324403579 | 0.00041415 |
| miR-344d-4p | −2.308193263 | 0.000742983 |
| miR-1964-3p | −2.267802775 | 0.00141537 |
| miR-320-3p | −2.212812155 | 0.00141537 |
| miR-128-3p | −2.031468208 | 0.003795561 |
| miR-351-3p | −1.963143876 | 0.004305251 |
| miR-1198-5p | −1.733517014 | 0.009437513 |
| miR-501-3p | −1.72302403 | 0.009643704 |
| miR-222-3p | −1.547695372 | 0.009966738 |
| let-7a-5p | −1.538094973 | 0.009966738 |
| miR-423-5p | −1.404879603 | 0.019014654 |

release of skeletal muscle-derived EVs are potentially beneficial in the treatment of various diseases and health management.

Our results showed that the intracellular $Ca^{2+}$ level significantly increased in the US-irradiated group, indicating that US irradiation promoted the uptake of $Ca^{2+}$ to myotubes. This result is supported by previous reports investigating $Ca^{2+}$ uptake by US in various types of cells (*Ambattu et al., 2020*; *Scheffer et al., 2014*; *Fan et al., 2010*; *Lentacker et al., 2014*; *Honda et al., 2004*). In the present study, the effect of US on $Ca^{2+}$ uptake was observed in an intensity-dependent manner. US irradiation has been reported to exert a physiological action to promote transient membrane permeabilization, which can promote the $Ca^{2+}$ influx pathways (*Fan et al., 2010*; *Cao et al., 2019*; *Zhou et al., 2008*), and the efficiency of permeabilization depends on the intensity of irradiation (*Kumon et al., 2009*; *Zeghimi et al., 2015*). Thus, the change of cell membrane structure is supposed to be the mechanism of the enhancement of $Ca^{2+}$ influx. In this study, EV release was promoted by US of 3.0 W/cm$^2$, and this effect was cancelled by inhibition of $Ca^{2+}$. Since intracellular $Ca^{2+}$ level is reported to be a factor that promotes EVs release (*Savina et al., 2003*; *Taylor et al., 2020*), US irradiation was shown to enhance EV release from myotubes via promotion of influx of $Ca^{2+}$.

We found that C2C12 myotube conditioned medium was capable of suppressing LPS-induced inflammatory responses in macrophages, and this effect was enhanced by US irradiation to myotubes. In addition, the anti-inflammatory effects were cancelled by eliminating EVs from the culture medium. This indicates that EVs in the culture medium were responsible for this effect, which is consistent with our previous study (*Yamaguchi et al., 2023*).

IL-1β is essential for an adequate acute inflammatory response to a variety of pathogens and injuries, but its excessive overexpression leads to sepsis, septic shock, or chronic inflammation (*Dinarello, 2009*; *Piccioli and Rubartelli, 2013*). IL-6 also contributes to host defense, however, dysregulated continual synthesis of it leads to prolonged chronic inflammation (*Tanaka et al., 2014*) and can drive tumorigenesis (*Chang et al., 2014*). Since blockade of these factors is reported to alleviate several inflammatory diseases (*Chevalier et al., 2005*; *Emsley et al., 2005*; *Choy et al., 2002*; *Nishimoto et al., 2004*; *Ito et al., 2004*; *Yokota et al., 2005*), eliciting the anti-inflammatory effects of skeletal muscle-derived EVs by US and suppressing the production of IL-1β and IL-6 suggests a new therapeutic strategy against inflammatory diseases utilizing US.

Meanwhile, it has been reported that the contents of EVs released from cells vary depending on the cellular microenvironment and that certain factors are selectively internalized in response to specific stimuli. The results of our analysis on miRNAs in EVs revealed that the EVs from US-exposed myotubes contain several unique miRNAs. However, only about 0.01% of the total miRNAs were altered by US irradiation, suggesting that the change was very small. Furthermore, no upregulation of the anti-inflammatory effect of myotube EVs by US was observed when the EV concentrations were equalized. Thus, the enhancement of the anti-inflammatory effect of myotube conditioned medium by US irradiation is assumed to be due to changes in the amount of EVs, not to changes in the content of EVs. Consistent with our results, a previous study reported that MSC-EVs regulated inflammatory responses in macrophages concentration dependently (*Kim et al., 2019*).

Analysis on miRNA profiles in the EVs revealed that the two most abundant miRNAs, miR-206-3p and miR-378a-3p, were common to both control and US EVs, and these two types accounted for more than 60% of the total miRNA contents in the both conditions. The most abundant miR-206 is a member of the skeletal muscle-specific myo-miR family of miRNAs (*Carpi et al., 2020*). It is reported that miR-206 targets IL-17A and REG3A and suppresses macrophage inflammation (*Huang et al., 2020*). Furthermore, Lin et al. reported that transfection of miR-mimic-206-3p into macrophages suppressed macrophage inflammation by targeting PPP3CA and transfection of miR-inhibitor-206-3p increased the level of inflammatory factors in macrophages (*Lin et al., 2020*). miR-378a is highly expressed in skeletal muscle and is involved in metabolism and mitochondrial energy homeostasis (*Krist et al., 2015*). Rückerl et al. identified miR-378a-3p as a factor contributing to the induction of anti-inflammatory macrophage reprogramming through IL-4-induced gene transcription by targeting Akt (*Rückerl et al., 2012*). In addition, Kris et al. reported that miR-378a has anti-inflammatory effects on macrophages and its deficiency enhances severity of inflammation (*Krist et al., 2020*). Based on these previous studies, it is assumed that myotube-derived EVs elicited the anti-inflammatory effects in macrophages by delivering these miRNAs. Additionally, miR-206 has been reported to modulate fat metabolism in diabetes (*Wu et al., 2017*) and control the tumorigenesis process of cancer (*Ding*

*et al., 2017*), and miR-378a has been shown to inhibit cardiac hypertrophy (*Ganesan et al., 2013*) and cancer growth (*Zeng et al., 2017*) and alleviate spinal cord injury (*Zhang et al., 2021*), indicating that these miRNAs are expected to have therapeutic effects against various diseases. Therefore, promoting the release of EVs containing these factors by US stimulation to skeletal muscle is potentially effective in the prevention and treatment of various diseases, and further investigations of its effects on other pathological conditions are expected.

On the other hand, while this study attributed the anti-inflammatory effect of US on skeletal muscle-derived EVs to increased EV release, we identified several miRNAs increased in US-induced EVs. Further studies are needed on this point for a more detailed understanding about the effect of US on skeletal muscle.

In summary, this study showed that US irradiation promoted the secretion of myotube-derived EVs which have anti-inflammatory effects and suggested that US irradiation to skeletal muscles is a potent candidate as a novel treatment for various inflammatory disorders.

## Materials and methods

### Cell culture

C2C12 myoblasts were purchased from the American Type Culture Collection (ATCC, USA). Myoblasts were cultured on 35 mm dishes (Iwaki) in Dulbecco's modified Eagle medium (DMEM, Wako Junyaku Co., Ltd., Japan) supplemented with 10% fetal bovine serum (FBS) at 37°C under 5% $CO_2$. At 90% confluence, differentiation into myotubes was initiated by changing the growth medium to differentiation medium (DMEM supplemented with 2% horse serum). Differentiation was continued for 6–7 days, refreshing the medium every 2 days. Efficiency of differentiation was confirmed by observing the

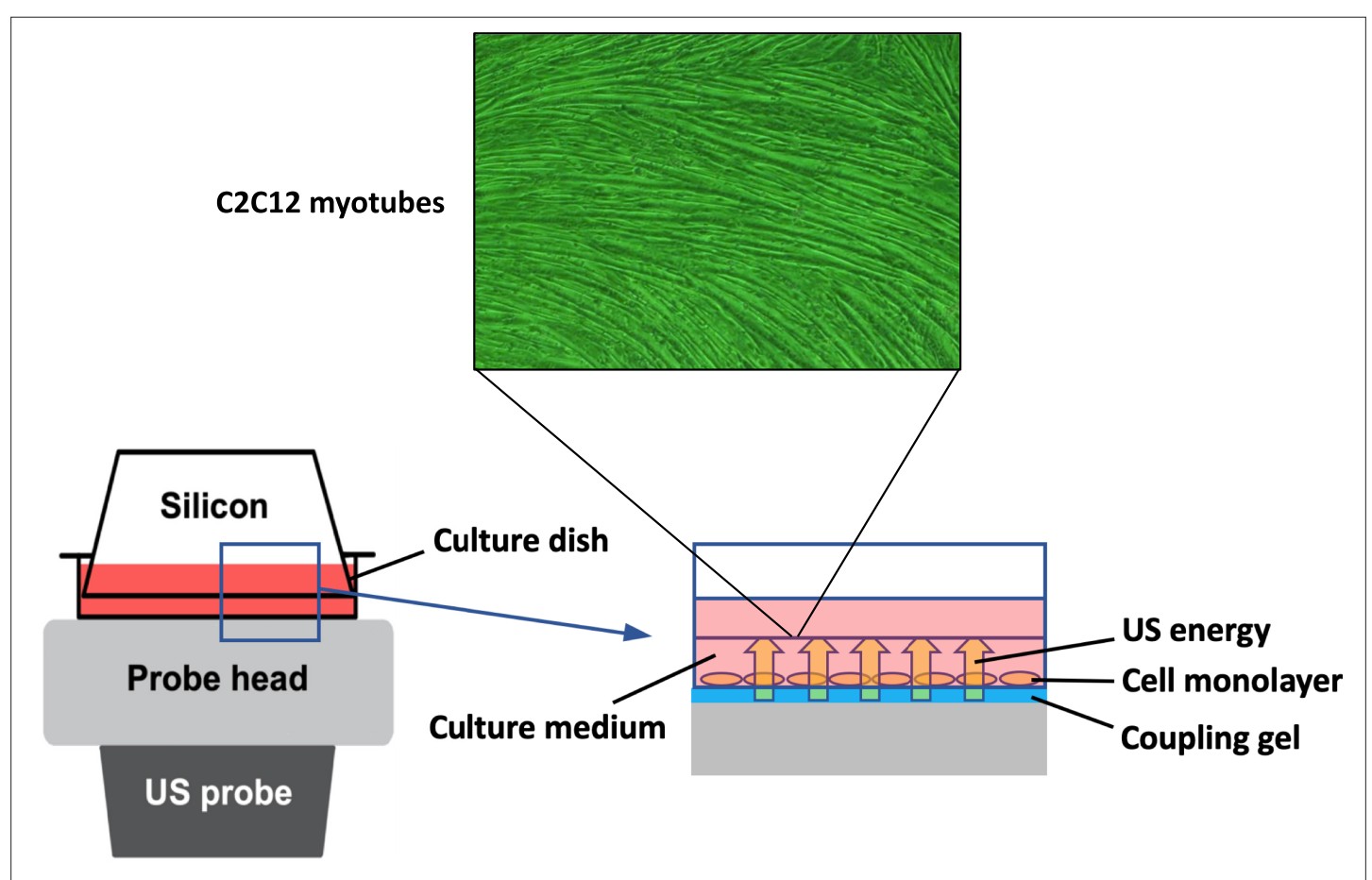

**Figure 6.** In vitro ultrasound (US) irradiation system. The culture dish was placed on the probe of a US transducer (SONICTIZER SZ-100, Minato Medical Science Co., Ltd., Japan). US waves are sent out from the probe placed under the culture dish, and surplus energy is absorbed by the silicon.

contractile ability using electrical stimulation (*Supplementary file 2*). US was applied to the myotubes for 5 min after changing the differentiation medium to serum-free medium. After incubation for 12 hr, the culture medium was collected to extract EVs.

To generate bone marrow-derived macrophages (BMDMs), bone marrow cells from femur and tibias of 7-week-old male C57BL/6 mice were harvested and cultured in Petri dishes with RPMI 1640 (DMEM, Wako Junyaku Co., Ltd., Japan) supplemented with 10% FBS, 1% penicillin/streptomycin, 1% L-glutamine, and 25% L929 cell supernatant for 8 days. Differentiated macrophages were harvested and plated in 12-well tissue culture plates with macrophage culture media (RPMI 1640 supplemented with 10% FBS, 1% penicillin/streptomycin, 1% L-glutamine, and 10% L929 cell supernatant) for subsequent experiments.

## US irradiation

To expose the myotubes to US, the culture dish was placed on the probe of an US transducer (SONIC-TIZER SZ-100, Minato Medical Science Co., Ltd., Japan). Coupling gel was applied to fill the space between the probe surface and the bottom of the dish and sterilized silicone was inserted into the culture medium and fixed to the dish with a distance of 2 mm from the bottom of the dish (*Figure 6*). The US parameters used were as follows: duty cycle of 20%; acoustic frequency of 1 MHz; duration of 5 min; repetition frequency of 100 Hz; beam nonuniformity ratio of 2.0; and effective radiation area of 8.0 cm$^2$ (*Maeshige et al., 2021*). In this study, three exposure intensities of 1.0 W/cm$^2$, 2.0 W/cm$^2$, and 3.0 W/cm$^2$ were investigated. After US irradiation, the myotubes were incubated at 37°C for 12 hr. We adopted the intensities of 1.0–3.0 W/cm$^2$ because output intensities of 0.1–2.5 W/cm$^2$ are typically applied for therapeutic purposes in clinical practice (*Draper, 2014*). After US irradiation, we monitored the temperature of the culture medium with a thermometer (TM-947SDJ, SATO-SHOJI, Japan) and confirmed it below 37°C to distinguish the advantage of US therapy from its thermal effect.

To investigate the involvement of intracellular Ca$^{2+}$ in the mechanism of EV release from myotubes, Ca$^{2+}$-free medium (Nacalai Tesque Inc, Japan) was used. The culture medium for Ca$^{2+}$-free groups was changed from normal medium (serum-free DMEM with calcium) to Ca$^{2+}$-free medium (serum-free DMEM without calcium) 1 hr before US irradiation.

## EV isolation

EVs were isolated from the conditioned medium of cultured myotubes with polymer precipitation as previously reported (*Ter-Ovanesyan et al., 2021*). The conditioned medium was collected and centrifuged for 15 min at 3000×*g* to remove cell debris. The supernatant was transferred to a sterile vessel and added Exo Quick-TC (System Biosciences, Palo Alto, CA, USA) (supernatant: Exo Quick reagent = 5:1). The tubes were stored overnight at 4°C and centrifuged at 1500×*g* for 30 min at 4°C. All traces of fluid were removed and the pellet was resuspended with 100 µL of PBS.

## EV characterization

A tunable resistive pulse sensing technology (qNano, IZON system; Izon Science Ltd., Christchurch, New Zealand) was used to measure the concentration, size distribution, and diameter of extracted EVs. The system was calibrated for voltage, stretch, pressure, and baseline current using standard beads: CPC100 (concentration; 1.0×10$^{10}$ beads/mL). An NP100 nanopore (for 50–200 nm size range) was used and data analysis was performed by a qNano IZON software.

## Macrophage treatment with conditioned medium of C2C12 myotubes

At 3 hr after US irradiation, the conditioned media were collected and centrifuged (1500×*g*, 10 min). Obtained supernatant was supplemented with 10% FBS, 1% penicillin/streptomycin, 1% L-glutamine, and 10% L929 cell supernatant and applied to BMDMs. After 1.5 hr treatment by myotube conditioned medium, BMDMs were incubated with macrophage culture media overnight (12 hr). Subsequently, BMDMs were treated with 100 ng/mL LPS for 1.5 hr for pro-inflammatory marker quantification. To clarify the involvement of EVs in the anti-inflammatory effects on macrophages, EV-depleted culture medium was added to macrophages. EV depletion was performed using Exo Quick-TC reagent as previously reported (*Mathew et al., 2019*).

## MTT assay

MTT assay was performed to evaluate cell viability 24 hr after US irradiation to myotubes. BMDM viability was also assessed 24 hr after EV treatment to examine the effect of the treatment with C2C12

conditioned medium on macrophages. The cells were incubated for 3 hr at 37°C with MTT solution (MTT, Wako Junyaku Co., Ltd., Japan) dissolved in culture medium at 0.5 mg/mL, then dissolved in DMSO. The absorbance at 595 nm was measured using a microplate reader.

## Zombie Red immunostaining

C2C12 myotube viability was assessed at 24 hr after US irradiation or treatment with 1% povidone-iodine (positive control) by Zombie Red immunostaining. This reagent is an amine-reactive fluorescent dye that is non-permeant to live cells but permeant to cells with a compromised plasma membrane. Myotubes were washed twice with PBS, stained with Zombie Red (1 : 1000) for 15 min at 37°C, and fixed with 4% paraformaldehyde for 30 min. Nuclei were stained with DAPI for 5 min at 37°C. The images were captured with a 10× objective on a BX50 (OLYMPUS, Japan).

## Total protein assay

Total protein content in myotubes was measured at 24 hr after US irradiation. The cells were washed with PBS and collected into tubes using a cell scraper. After centrifugation at 13,000 rpm for 20 s, the pellet was processed with 100 µL of PRO- PREP reagent (iNtRON Biotechnology Co., Ltd. Japan), followed by incubation on ice for 20 min. After centrifugation at 13,000 rpm for 5 min, the protein concentration was analyzed using the Bradford method.

## CS activity

Energy metabolism was evaluated 24 hr after US irradiation to the myotube by CS assay as previously reported (*Srere, 1963*). Briefly, the samples of total protein extraction were mixed with the following solution: distilled water (DW)+3mMAcetylCoA+1mMDTNB (1 mM DTNB+1 M Tris-HCL). Oxaloacetate was added to the mixture and the absorbance at 415 nm was measured using a microplate reader every 2 min for 10 min at 37°C.

## Intracellular Ca$^{2+}$ levels

The concentration of intracellular Ca$^{2+}$ in the medium of C2C12 myotubes was measured using a Metallo assay calcium kit LS (CPZIII) (Metallogenics Ltd., Japan). This method determines intracellular Ca$^{2+}$ levels by observing the coloration in the visible region caused by the chelate complex formation between chlrophosphonazo-III (CPZIII) and Ca$^{2+}$. Briefly, the cells were collected and centrifuged at 13,000 rpm for 20 s to remove the supernatant. The pellet was suspended with a mixture of 200 µL of RIPA buffer (RIPA Lysis Buffer, SCB Ltd., Japan) and 1 µL of hydrochloric acid (2 M). After incubation for 30 min, the samples were centrifuged at 10,000 rpm for 10 min and the Ca$^{2+}$ concentration in the supernatant was analyzed. Ca$^{2+}$ was dissociated by low pH and collected as the supernatant. Finally, the absorbance was converted to concentration (mM) according to the manufacturer's protocol.

## Quantitative real-time PCR

To measure gene expression in BMDMs, mRNA was isolated with TRIzol RNA Isolation protocol and used to make cDNA using iScript cDNA Synthesis Kit (Bio-Rad). The StepOne Real-Time PCR System was used to analyze the samples under the following conditions: 95°C (3 min), 40 cycles of 95°C (10 s), and 60°C (30 s). The reaction mixture consisted of 8 µL cDNA, 1.5 µL 10× buffer, 0.3 µL 10 mM dTNPs, 1.5 µL 5 µM primers for each gene used in the study (F+R), 3.58 µL H$_2$O, 0.075 µL Go Taq DNA polymerase and 0.045 µL 2×SYBR green (Invitrogen). Target genes were the pro-inflammatory markers *Il-1b* and *Il-6* (*Moore et al., 2013*). Relative expression values for target genes were calculated by normalization to the expression of glyceraldehyde-3-phosphate dehydrogenase (*Gapdh*). Data was analyzed using the ΔΔCT method. Sequences for qPCR primers are shown in ***Supplementary file 3***.

## miRNA sequencing

miRNA was extracted from myotube-derived EVs using TRIzol reagent (Takara Biotechnology, Japan) according to the manufacturer's instructions. Raw miRNA sequence data were obtained using an Illumina NovaSeq 6000 machine. After acquiring the raw data, the fold change (mean of each miRNA in the US group/mean of each RNA in the control group) and p-values were calculated for each miRNA. These p-values were used to calculate the false discovery rate (FDR) for each miRNA, which was

further used as a filter to identify significant miRNAs with a fold change ≥2 or ≤0.5 and an FDR <0.05. The volcano plots were generated using the R 3.5.3 software.

## Statistical analysis

Statistical analysis was conducted using Statistical 4 (OMS, Tokyo, Japan). For two-group comparison, Student's t-test was used and for multiple comparisons, ANOVA (Tukey's multiple comparison test as post hoc) was used. Power analysis using G Power software (*Kang, 2021*) was conducted to determine the sample size for each experiment to provide a power of at least 0.8 at a significance level of 0.05 ($\alpha=0.05$, $\beta=0.2$).

## Acknowledgements

This study was supported by JSPS KAKENHI (grant number 17H04747 and 21H03852). We are grateful to Dr. P Kent Langston (Department of Immunology, Harvard Medical School and Evergrande Center for Immunologic Diseases, Harvard Medical School, Boston, MA, USA) for providing insightful advice.

## Additional information

### Funding

| Funder | Grant reference number | Author |
| --- | --- | --- |
| Japan Society for the Promotion of Science | 17H04747 | Noriaki Maeshige |
| Japan Society for the Promotion of Science | 21H03852 | Noriaki Maeshige |

The funders had no role in study design, data collection and interpretation, or the decision to submit the work for publication.

### Author contributions

Atomu Yamaguchi, Conceptualization, Validation, Investigation, Writing – original draft; Noriaki Maeshige, Conceptualization, Resources, Supervision, Funding acquisition, Methodology, Project administration, Writing – review and editing; Hikari Noguchi, Data curation, Formal analysis, Visualization; Jiawei Yan, Supervision, Project administration, Writing – review and editing; Xiaoqi Ma, Data curation, Formal analysis, Writing – review and editing; Mikiko Uemura, Validation, Investigation, Visualization, Methodology, Writing – review and editing; Dongming Su, Conceptualization, Supervision, Methodology, Writing – review and editing; Hiroyo Kondo, Conceptualization, Supervision, Writing – review and editing; Kristopher Sarosiek, Supervision, Writing – original draft, Project administration, Writing – review and editing; Hidemi Fujino, Conceptualization, Supervision, Funding acquisition, Methodology, Writing – review and editing

### Author ORCIDs

Noriaki Maeshige (iD) https://orcid.org/0000-0002-3573-347X

### Ethics

The animal experimentation was conducted according to the protocol reviewed and approved by the Institutional Animal Care and Use Committee of Kobe University (Permit No. P210803).

Reviewer #1 (Public Review): https://doi.org/10.7554/eLife.89512.3.sa1
Reviewer #2 (Public Review): https://doi.org/10.7554/eLife.89512.3.sa2
Author Response https://doi.org/10.7554/eLife.89512.3.sa3

## Additional files

### Supplementary files

• Supplementary file 1. Lists of specific miRNAs in ultrasound (US)-treated/untreated groups. miRNA-sequencing analysis in extracellular vesicles from US-treated/untreated C2C12 myotubes was conducted.

• Supplementary file 2. Myotube contraction by electrical stimulation. Myotubes were electrically stimulated (30 mA at 1 Hz for 15 ms at 985 ms intervals) with an electrical pulse generator (ITO Co., Ltd, Saitama, Japan) to confirm differentiation. Separately plated cells were used to confirm contraction and the cells used for the experiments were not electrically stimulated.

• Supplementary file 3. Sequences for qPCR primers used in this study.

• MDAR checklist

### Data availability

The RNA-seq data can be found in NCBI database (BioProject ID: PRJNA1044751).

The following dataset was generated:

| Author(s) | Year | Dataset title | Dataset URL | Database and Identifier |
|---|---|---|---|---|
| Maeshige N | 2023 | Ultrasound-induced extracellular vesicles from C2C12 myotubes | https://www.ncbi.nlm.nih.gov/bioproject/?term=PRJNA1044751 | NCBI BioProject, PRJNA1044751 |

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
