## [Editor Report · eLife assessment]

This study illuminates the effects of ultrasound-induced extracellular vesicle interactions with macrophages. It provides **solid** data offering insights that will be potentially **useful** in exploring therapeutic approaches to inflammation modulation, by suggesting that ultrasound-treated myotube vesicles can suppress macrophage inflammatory responses.

---

## [Referee Report · Reviewer #1 (Public Review)]

Summary:

The authors embarked on a journey to understand the mechanisms and intensity-dependency of ultrasound (US)-induced extracellular vesicle (EV) release from myotubes and the potential anti-inflammatory effects of these EVs on macrophages. This study builds on their prior work from 2021 that initially reported US-induced EV secretion.

Strengths:

1. The finding that US-treated myotube EVs can suppress macrophage inflammatory responses is particularly intriguing, hinting at potential therapeutic avenues in inflammation modulation.

Weaknesses:

1. The exploration of output parameters for US induction appears limited, with only three different output powers (intensities) tested, thus narrowing the scope of their findings.

2. Their claim of elucidating mechanisms seems to be only partially met, with a predominant focus on the correlation between calcium responses and EV release.

3. While the intracellular calcium response is a dynamic activity, the method used to measure it could risk a loss of kinetic information.

4. The inclusion of miRNA sequencing is commendable; however, the interpretation of this data fails to draw clear conclusions, diminishing the impact of this segment.

While the authors have shown the anti-inflammatory effects of US-induced EVs on macrophages, there are gaps in the comprehensive understanding of the mechanisms underlying US-induced EV release. Certain aspects, like the calcium response and the utility of miRNA sequencing, were not fully explored to their potential. Therefore, while the study establishes some findings, it leaves other aspects only partially substantiated.

---

## [Referee Report · Reviewer #2 (Public Review)]

Summary:

The work demonstrates that high-intensity ultrasound produces a release of extracellular vesicles from murine myotubes that is dependent on ultrasound intensity. It shows that this increase in extracellular vesicles is abolished in a nominally zero Ca2+ solution. It then shows that these vesicles reduce the mRNA levels of IL-1b and IL-6 in murine bone marrow-derived macrophages and uses a dilution technique to demonstrate that the number, but not the type of vesicles is responsible for this change in mRNA expression. It also compares the miRNA levels in constitutively-released vesicles with those released by high-intensity ultrasound.

Strengths:

The experiments are logically sequenced. It is very helpful to see assessment of the viability of the preparations.

The results are presented fairly clearly and the statistical approach is described. The findings are reasonably clear and the writing is succinct.

Weaknesses:

The work is quite limited in scope and of limited novelty, largely recapitulating work from the first-named author's own recent publications.

Thus, perhaps the most significant weakness of this study is that it makes claims of mechanisms, or of clinical or therapeutic relevance, that are not supported or even addressed by the study.

The aspects of the current work which are novel are hard to identify because the statement of aims is too broad and therefore encapsulates previous work. In addition, the introduction and discussion are vague and fail, for example, to mention the cell types used in the previous studies that are quoted. This means that it is not obvious from the Introduction whether the present study is at all novel.

The size of the study is quite small, with most experiments employing n = 4. This inevitably means that, for example, there is no significant effect of the lower power levels of ultrasound despite shifts in the mean values that might be of interest. Thus, the study appears underpowered. This problem is compounded by a failure to use appropriate analysis methods in the studies looking at dose-responses, where a regression analysis might be more appropriate than multiple individual t-tests / ANOVA.

The assessment of the role of Ca2+ is important but incomplete. Measurement of whole-cell Ca2+ levels is not really a substitute for measuring cytosolic Ca2+ as cell volume changes and sarcoplasmic reticulum Ca2+ changes would greatly influence the possible meaning of the findings. Furthermore, a statement that Ca2+ increase causes the vesicle release could only be supported by experiments that increase intracellular Ca2+, such as the use of a Ca2+ ionophore.

mRNA expression levels of IL-1b and IL-6 are reported. There should also be a report of a non-inflammatory mRNA to act as a control.

---

## [Author Response]

The following is the authors’ response to the original reviews.

**Reviewer #1 (Public Review):**
While the manuscript was reasonably clearly written and the methodology and results sound, it is not clear what the real contribution of the work is. The authors' findings - that ultrasonic stimulation is capable of altering intracellular Ca2+ to effect an increase in EV secretion from cells as long as the irradiation does not affect cell viability-is well established (see, for example, Ambattu et al., Commun Biol 3, 553, 2020; Deng et al., Theranostics, 11, 9 2021; Li et al., Cell Mol Biol Lett 28, 9, 2023). Moreover, the authors' own work (Maeshige et al., Ultrasonics 110, 106243, 2021) using the exact same stimulation (including the same parameters, i.e., intensity and frequency) and cells (C2C12 skeletal myotubes) reported this. Similarly, the authors themselves reported that EV secretion from C2C12 myotubes has the ability to regulate macrophage inflammatory response (Yamaguchi et al., Front Immunol 14, 1099799, 2023). It would then stand to reason that a reasonable and logical deduction from both studies is that the ultrasonic stimulation would lead to the same attenuation of inflammatory response in macrophages through enhanced secretion of EVs from the myotubes.

We appreciate your comments and suggestions. Ambattu et al. in their report stated that the high frequency acoustic stimulation they used has a less effect on cell membranes than the 1 MHz ultrasound that we used in this study. Deng et al. and Li et al. applied low intensity pulsed ultrasound (LIPUS) (about 300 mW/cm2) in their studies. In this study, we assumed that ultrasound induced increase in EV secretion via increased Ca2+ influx into the cell by enhancing cell membrane permeability, and since it has been reported that the effect of ultrasound-induced enhancement in cell membrane permeability increases in an intensity-dependent manner (Zeghimi et al., 2015), we applied intensities of 1-3 W/cm2. While previous studies using LIPUS have used 15 minutes of irradiation, the high intensity employed in this study was capable to promote EV release after 5 minutes of stimulation. We have added the above explanation to the introduction in the revised version of the manuscript. Furthermore, while the previous studies used other types of cells, the main purpose of this study was to determine the optimal ultrasound intensity to promote EV release from skeletal muscle and to determine whether ultrasound-induced EVs are qualitatively altered compared to those released under normal conditions, thereby validating the anti-inflammatory effects of ultrasound-induced muscle EVs. Our previous study (Maeshige et al. 2021) used the same muscle cells but did not investigate an intensity dependence, so this is the first study to show that ultrasound irradiation promotes EV release in an intensity-dependent manner in muscle. In addition, we would like to emphasize that this study also goes beyond our previous study in the method of stimulation. Specifically, the present study a more efficient 5-minute irradiation protocol was used, whereas the previous study have adopted a 9-minute intervention.

We understand that the results of this study are predictable from two of our previous studies, but since stimulus-induced EVs may be qualitatively different compared to EVs released under normal conditions (Kawanishi et al., 2023; Li et al., 2023), it is worthwhile to examine the effects of stimulus-induced EVs. This explanation has been added in the introduction of revised version of the manuscript.

The authors' claim that 'the role of Ca2+ in ultrasound-induced EV release and its intensity-dependency are still unclear', and that the aim of the present work is to clarify the mechanism, is somewhat overstated. That ultrasonic stimulation alters intracellular Ca2+ to lead to EV release, therefore establishing their interdependency and hence demonstrating the mechanism by which EV secretion is enhanced by the ultrasonic stimulation, was detailed in Ambattu et al., Commun Biol 3, 553, 2020. While this was carried out at a slightly higher frequency (10 MHz) and slightly different form of ultrasonic stimulation, the same authors have appeared to since establish that a universal mechanism of transduction across an entire range of frequencies and stimuli (Ambattu, Biophysics Rev 4, 021301, 2023).

In this study, we showed that Ca2+ is involved in ultrasound-induced EV release using Ca2+-depleted culture medium, but since we did not examine the mechanism in more detail than that, we modified the introduction to avoid overstating.

Similarly, the anti-inflammatory effects of EVs on macrophages have also been extensively reported (Li et al., J Nanobiotechnol 20, 38, 2022; Lo Sicco et al., Stem Cells Transl Med 6, 3, 2017; Hu et al., Acta Pharma Sin B 11, 6, 2021), including that by the authors themselves in a recent study on the same C2C12 myotubes (Yamaguchi et al., Front Immunol 14, 1099799, 2023). Moreover, the authors' stated aim for the present work - clarifying the mechanism of the anti-inflammatory effects of ultrasound-induced skeletal muscle-derived EVs on macrophages - appears to be somewhat redundant given that they simply repeated the microRNA profiling study they carried out in Yamaguchi et al., Front Immunol 14, 1099799, 2023. The only difference was that a fraction of the EVs (from identical cells) that they tested were now a consequence of the ultrasound stimulation they imposed.That the authors have found that their specific type of ultrasonic stimulation maintains this EV content (i.e., microRNA profile) is novel, although this, in itself, appears to be of little consequence to the overall objective of the work which was to show the suppression of macrophage pro-inflammatory response due to enhanced EV secretion under the ultrasonic irradiation since it was the anti-inflammatory effects were attributed to the increase in EV concentration and not their content.

In comparison with the current study, our previous study observed EVs secreted only from muscle in normal condition. However, the purpose of the current study is to answer the question whether ultrasound treatment could enhance the effect of EVs and change the encapsuled miRNAs. Although we identified several miRNAs which are specifically induced by ultrasound, further studies are needed to demonstrate the effect of those miRNAs derived from ultrasound-treated muscles on macrophages. We have mentioned this limitation in the discussion of the revised manuscript.

**Reviewer #1 (Recommendations For The Authors):**

This reviewer felt that there was a lack of novelty in the manuscript and that the results of the work confirm conclusions that could have been logically deduced from a combination of the authors' preceding work (Maeshige et al., Ultrasonics 110, 106243, 2021 and Yamaguchi et al., Front Immunol 14, 1099799, 2023). The contribution of the work could perhaps be elevated if the authors were to focus more on whether the 0.01% of altered miRNA has any impact on cellular activity.

As mentioned above, the present study is novel compared to our previous studies for examining the effects of ultrasound-induced EVs. In addition, the fact that EV content is maintained after ultrasound stimulation rather indicates that ultrasound can be used as a highly stable and effective method of promoting EV release.

A further, albeit more minor, recommendation is to omit lines 73-80 in the manuscript. The discussion on physical exercise for promoting EV secretion together with the non-invasive nature of ultrasound therapy is very misleading as it creates the impression that the authors' work can be applied as a direct intervention on a patient. This was not shown in the work, which was limited to irradiating cells ex vivo.

We agree and have edited the introduction.

**Reviewer #2 (Public Review):**
1. The exploration of output parameters for US induction appears limited, with only three different output powers (intensities) tested, thus narrowing the scope of their findings.

We appreciate your comments and suggestions. The intensity of LIPUS is basically in the ~0.3 W/cm2 range, and in clinical practice, ~2.5 W/cm2 is considered to be a safe intensity to irradiate the human body (Draper, 2014). Therefore, 3.0 W/cm2 is also a fairly high intensity for the human body, so 3.0 W/cm2 was set as the maximum intensity in this study.

1. Their claim of elucidating mechanisms seems to be only partially met, with a predominant focus on the correlation between calcium responses and EV release.

The focus of this study was to examine the effects of ultrasound-induced EVs on the inflammatory responses of macrophages and not on the detailed mechanism of calcium involvement. We revised the introduction to make the purpose of this study clearer.

1. While the intracellular calcium response is a dynamic activity, the method used to measure it could risk a loss of kinetic information.

Although we did not examine the kinetic action of calcium, we believe that Ca2+ is at least proven to be involved to the EV-promoting effect of ultrasound on muscle, since the enhancement of EV release by ultrasound was canceled by elimination of calcium from the culture medium. Furthermore, real-time measurement of Ca2+ after ultrasound irradiation has shown that ultrasound irradiation promotes Ca2+ influx into cells immediately after the irradiation. (Fan et al., 2010).

1. The inclusion of miRNA sequencing is commendable; however, the interpretation of this data fails to draw clear conclusions, diminishing the impact of this segment.

Although we identified several miRNAs which are specifically induced by ultrasound, further studies are needed to demonstrate the effect of those miRNAs derived from US-treated muscles on macrophages. We have mentioned this limitation in the discussion of the revised version of manuscript.

While the authors have shown the anti-inflammatory effects of US-induced EVs on macrophages, there are gaps in the comprehensive understanding of the mechanisms underlying US-induced EV release. Certain aspects, like the calcium response and the utility of miRNA sequencing, were not fully explored to their potential. Therefore, while the study establishes some findings, it leaves other aspects only partially substantiated.

As stated above, the main purpose of this study was to examine the effects of ultrasound-induced EVs on the inflammatory responses of macrophages. We set detailed investigation on the mechanism of ultrasound-induced EV release as our next step and have revised the introduction and discussion of the revised manuscript to make the purpose and limitation of this study clearer.

**Reviewer #2 (Recommendations For The Authors):**
The author's exploration into the role of Ca2+ in the context of US-induced EV release is a timely endeavor, especially given the growing interest in understanding the cellular dynamics associated with external stimulants like ultrasound. Nevertheless, the manuscript does not unambiguously define the mechanism of action and its broader implications.Ca2+ has long been established as a versatile intracellular messenger, governing a myriad of cellular processes. There is a wealth of methodologies, from specific inhibitors to specialized assays, tailored to dissect the role of Ca2+ in diverse contexts. In the specific case of US-induced Ca2+ activity, the expectation would be for a clear, mechanistic delineation of how this ionic surge drives EV release. Yet, this study stops short of providing those details. It is imperative for the authors to dig deeper, employing a diverse set of tools at their disposal, to fill this knowledge gap.

Recently, it was reported that increased Ca2+ influx causes an increase in EV secretion via the plasma membrane repair protein annexin A6 (Williams et al. 2023). However, the full mechanism of how an increase in intracellular Ca2+, let alone ultrasound-induced Ca2+, promotes EV release has not yet been understood yet, and it is beyond the scope of this study to elucidate this part of the mechanism.

Furthermore, the paper raises another important question: Which specific proteins are pivotal in orchestrating the US-induced Ca2+ entry in myotubes? Addressing this would not only enhance the manuscript's novelty but would also contribute a vital piece to the puzzle of understanding US-cellular interactions.

Ultrasound increases Ca2+ uptake by increasing cell membrane permeability by sonoporation, rather than via protein reactions (Fan et al., 2010). We added this explanation to the introduction in the revised version of manuscript.

Lastly, while the report touches upon the influence of varying US output power on EV concentrations, it piques curiosity about potential effects beyond the 3W/cm2 threshold. It's observed that cell viability isn't compromised at this intensity, suggesting room for further exploration. Would a higher intensity yield a proportionally increased EV release, or is there a saturation point? Conversely, could intensities beyond 3W/cm2 begin to have deleterious effects on the cells? These are crucial considerations that merit investigation to realize the full potential of US as a modulatory tool, both for research and therapeutic applications.

As mentioned above, 3.0 W/cm2 was adopted as the maximum intensity in this study with reference to the intensity used in clinical practice. In addition, since the cytotoxicity and therapeutic effects of ultrasound depend not only on intensity but also on other parameters such as duty cycle, acoustic frequency, pulse repetition frequency and duration, so a comprehensive analysis of the effects of ultrasound on cells at various parameter settings would be valuable as an independent study.